# Has the Volume-Based Drug Purchasing Approach Achieved Equilibrium among Various Stakeholders? Evidence from China

**DOI:** 10.3390/ijerph19074285

**Published:** 2022-04-03

**Authors:** Qian Xing, Wenxi Tang, Mingyang Li, Shuailong Li

**Affiliations:** 1Department of Public Administration, School of International Pharmaceutical Business, China Pharmaceutical University, Nanjing 211198, China; 3220040610@stu.cpu.edu.cn (Q.X.); 3321041321@stu.cpu.edu.cn (M.L.); 3221041344@stu.cpu.edu.cn (S.L.); 2Department of Pharmacoeconomics, Center for Pharmacoeconomics and Outcomes Research, China Pharmaceutical University, Nanjing 211198, China

**Keywords:** China, volume-based drug purchasing, stakeholder analysis, health impact assessment, multidimensional scaling, policy evaluation

## Abstract

Volume-based drug purchasing by China’s health insurance system currently represents the largest group purchasing organization worldwide. After exchanging the market that accounted for nearly half of the volume of the healthcare system for the ultra-low-price supply of limited drugs, what are the effects on patient and funding burdens, drug accessibility, and clinical efficacy? We aimed to verify the effectiveness of the policy, explore the reasons behind the problem and identify regulatory priorities and collaborative measures. We used literature and reported data from 2019 to 2021 to conduct a stakeholder analysis and health impact assessment, presenting the benefit and risk share for various dimensions. The analysis method was a multidimensional scaling model, which visualized problematic associations. Seventy-nine papers (61 publications and 18 other resources) were included in the study, with 22 effects and 36 problems identified. The results indicated favorable affordability and poor accessibility of drugs, as well as high risk of reduced drug quality and drug-use rationality. The drug-use demand of patients was guaranteed; the prescription rights of doctors regarding clinical drug use were limited; unreasonable evaluation indicators limited the transformation of public hospitals to value- and service-oriented organizations; the sustainability of health insurance funds and policy promotion were at risk; and innovation by pharmaceutical companies was accelerated. The problems associated with high co-occurrence frequencies were divided into the following clusters: cost control, drug accessibility, system rationality, policy fairness, drug quality, and moral hazards. These findings suggested that China has achieved short-term success in reducing the burden on patients and reducing fund expenditure. However, there were still deficiencies in guaranteed supply, quality control, and efficacy tracking. The study offers critical lessons for China and other low- and middle-income countries.

## 1. Introduction

Controlling rising drug costs is a global concern. Different countries, through either social or commercial insurance systems, have developed centralized procurement models to varying degrees, and have explored delivery mechanisms to provide the lowest possible drug prices through price–volume agreements [1]. Joint procurement of medicines is one way to improve access and fairness in low- and middle-income countries [2]. Health policy bodies such as the World Health Organization (WHO) have adopted policies such as strategic purchasing and centralized procurement to provide fairness and reform access to pharmaceutical systems in developing countries. They see centralized procurement as an important tool for sustainable supply improvement and financial access, and to ensure the safety and efficacy of essential medicines in these countries [3]. In China, the government has made great progress in establishing a universal medical insurance system. China’s health insurance system, as the largest payer and purchaser, aims to provide health services and financial protection for the public [4]. The national health insurance system covered 1.361 billion people with a fund size of 496 billion USD in 2020 [5]. In recent years, increased government support for innovation in the pharmaceutical industry has seen increasing numbers of high-value innovative drugs included in the health insurance system, while the pressure points in terms of fund outflows have also increased. In order not to interfere with clinical needs, the National Healthcare Security Administration of China has accelerated the introduction of other measures ensuring the sustainability of fund payments under low-level financing requirements [4]. Among these, centralized drug procurement, as a governance tool aimed at controlling costs and guaranteeing the quality of drugs, was mentioned for the first time at national policy level [6]. The coverage focuses on drugs with high usage and procurement within the basic health insurance drug catalogue [7]. This policy aims to obtain preferential drug prices by integrating national market purchases, and is termed national centralized drug volume-based procurement (VBP). It was piloted in 11 cities in January 2019 and then promoted throughout China eight months later [8].

By the end of 2019, the size of the drug end market had reached 287.3 billion USD, of which generic drugs accounted for more than 160 billion USD, while innovative drugs accounted for less than 16 billion USD [9]. The number of generic drug approvals amounted to ~95% of all drug approvals, and the price of generic drugs was 20–90% lower than the price of corresponding original drugs. Thus, generic drugs have become the mainstay of the pharmaceutical consumer market in China [9]. In the past, low-quality production of generic drugs was a serious problem, and approval standards were low [10]. With the revision of China’s drug administration laws and the full implementation of the drug listing licensee system, emphasis has been placed on the regulation of the entire life cycle of drugs, thus better safeguarding the quality of drugs [11]. In 2016, the General Office of the Chinese State Council released the Generic Drug Quality Consistency Evaluation guidelines, which set out programmatic regulations for evaluations, time limits, methods, and reference preparations [12]. The active ingredients and in vivo bioequivalence of generic drugs were required to be consistent with those of the original drugs [13]. This overcame the previous loopholes in generic drug supervision and interrupted the disorderly homogeneous competition that had arisen, and qualified these generic drugs for inclusion in VBP. The general public’s view of VBP as being mainly aimed at cost control is too shallow [14]. The goal of the central government’s policy is to target the national health system and integrate its purchasing power to effectively reduce patient burden in relation to drug use, promoting the reform of public institutions, leveraging the benefits of the “three-medicine” (medical services, health insurance, and medicine) mechanism, improving the healthcare market, and cleaning up healthcare ecology.

Five rounds of VBP have been conducted in China. In January 2019, four municipalities and seven key cities were selected as pilot locations, and thus the policy is also known as “4 + 7” [15]. At this stage, the positive effect of “quantity for price” first appeared. Therefore, the next four rounds of VBP were carried out in January 2020, August 2020, February 2021, and June 2021, respectively. A total of 218 drugs were covered, including common therapeutic drugs such as those for hypertension, diabetes mellitus, gastrointestinal diseases, and infections, as well as drugs for major diseases such as antitumor drugs, immunomodulators, and psychiatric drugs. A wide range of people benefited, and the mean price reduction reached 54% [16]. Data released by the National Healthcare Security Administration showed that the first four rounds of VBP involved 157 drugs and saved health insurance funds 160 billion USD [17]. Currently, VBP is expanding into the fields of biological medicines, Chinese patent medicines, and consumable equipment [18]. Faced with the strong impact of the COVID-19 pandemic, all provinces implemented the policies on time, except for Wuhan Province which deferred the second batch of procurement, and VBP began pilot procurement of nucleic acid testing reagents and supporting consumables [19].

VBP is organized at the national level. The purchaser (public hospital or government-run primary healthcare institution) agrees on the purchase volume, submits the estimated annual purchase volume based on drug consumption in the previous year, and guarantees sufficient purchase quantities. The drug supplier bids or bargains through the centralized procurement platform, determines the final purchase price, and enters the procurement transaction mode [20,21]. Indeed, a number of national and international allied organizations have implemented centralized procurement as a means of creating economies of scale, increasing purchasing power and reducing health system costs [22,23,24]. For example, the United Nations International Children’s Emergency Fund (UNICEF) has implemented a centralized banded procurement program, and by consolidating vaccine procurement across multiple low-income countries, UNICEF has succeeded in reducing the purchase price of vaccines and lowering the cost of participation for vaccine companies through standardization of procurement processes [25,26]. In addition, other countries such as the Gulf Cooperation Council, the East African Community, Delhi, India, Brazil, Brazil, the Caribbean and Mexico are engaged in volume purchasing with the aim of addressing high drug prices and poor access to essential medicines [27,28,29]. Although these countries differ in terms of the type of procurement, lead sector, participants and process, all have contributed to increasing access to medicines. In addition to the banded procurement approach, Spain has ensured the availability of essential medicines with high clinical use through the signing of discounted generic contracts, and the UK through separate bidding for generics [30,31].

VBP in China is based on the principle of group purchasing organizations (GPOs). The GPO is a product of medical institutions’ initiatives to save costs and promote fine-tuned management. GPOs are mainly engaged in competitive bidding and supply chain management for drug procurement, driving effective service provision through economies of scale and improved bargaining power [32]. The GPO model is not perfect, and problems such as price competition and increasing coordination costs (e.g., inventory, transport, supervision costs) have occurred in other countries including the United States and Singapore [33]. Additionally, limited evidence suggests that the charging practices of some GPO suppliers have led to problems in the supply chain and shortages of some drugs [34]. During the pilot round in China, in Shenzhen and Shanghai, problems emerged such as monopolistic competition pressure due to administrative intervention, conflict with existing policies, and lack of a suitable supervisory system [35]. Given its wide coverage and rapid progress, VBP in China is currently experiencing similar dilemmas.

It is difficult to conduct a general evaluation of the effects and problems of VBP. What is the impact on patients, fund burdens, drug accessibility and clinical efficacy? Has it reconciled the interests, values and attitudes of policy implementers and policy recipients? Does it provide experience and lessons for other low- and middle-income countries to carry out health policy reforms? Most of the existing studies have analyzed the pros and cons from the perspective of a particular stakeholder, or as a game between two players, and more often than not, they have used procurement data from selected regions to conduct empirical studies on the effects of cost control. This study takes the perspective of universal health and sustainable development of the policy, and aims to synthesize as completely as possible the existing qualitative and quantitative evaluation results, to systematically analyze the positive impacts, obstacles and risks of the policy, and to a certain extent fill the gaps in the current research into and analysis of the effects of VBP. By comparing the findings with the intended effects, we identify the supporting measures that need to be improved and the regulatory priorities that need to be strengthened in the subsequent promotion process, to assist the problems faced during health system reform in terms of drug quality risks and supply shortages, and to provide new ideas for other developing countries to optimize the allocation of health resources.

## 2. Materials and Methods

### 2.1. Data Sources

#### 2.1.1. Publications

A literature search of the following Chinese and English databases was conducted: China National Knowledge Infrastructure (https://www.cnki.net, accessed on 12 October 2021), Wanfang Database (https://www.wanfangdata.com.cn, accessed on 12 October 2021), Medline (via Pubmed) (https://pubmed.ncbi.nlm.nih.gov, accessed on 12 October 2021), and Web of Science (https://www.webofknowledge.com, accessed on 12 October 2021). Chinese search terms were based on the topic “药品集中采购 (centralized drug procurement)” + “带量采购 (volume-based procurement)” + “4 + 7,” and English search terms were based on titles/abstracts containing the terms “centralized drug procurement” OR “centralized drug purchasing” OR “volume-based procurement” OR “volume-based purchasing” OR “‘4 + 7′ policy.” The time frame for the search spanned the VBP planning period to the most recent publication date, i.e., 1 November 2018–10 October 2021.

The inclusion criteria were as follows: (1) the current status, effects, and problems of China’s centralized drug procurement were mentioned explicitly; and (2) the study objects included one or more of the subjects of health insurance management, hospital management, medical workers, patients, pharmacies, and pharmaceutical companies. The exclusion criteria were as follows: (1) prefecture-level VBP; (2) the research topic and content were clearly not in compliance, such as VBP of healthcare consumables, enterprise innovation strategies, the health insurance management model, and group procurement of drugs; (3) the focus was on the implementation of centralized VBP as well as bidding and pricing processes, rather than on effect evaluation; (4) empirical research based on data collected before centralized procurement commenced in the pilot areas in November 2018; (5) the publication types were meeting notices, guidelines, submission instructions, column introductions, or journal catalogs; (6) reprinted documents and repeat publications; and (7) inability to access the full text.

#### 2.1.2. Other Types of Data

The literature involved certain publication lags and biases. Therefore, in this study, we also included as supplementary literature official media reports, authoritative official medical accounts, and academic seminars. The search strategy and inclusion/exclusion criteria were in accordance with those used for the literature search. The time frame for the search was 1 January 2021–10 October 2021. To guarantee the quality of the content, the selected official service accounts were those with stable volumes of readership and popularity.

Second, we included government policy documents. To objectively present the goals and key points of the policy design, notices about drug purchasing (excluding policy interpretations and briefings) were collected from the government’s official websites, such as those of the General Office of the State Council and the National Healthcare Security Administration of the People’s Republic of China. The time frame for the search was 1 November 2018–10 October 2021.

### 2.2. Data Extraction

Initial screening of the title and abstract was conducted by two independent raters (M.L. and S.L.). Both independent raters reviewed full-text versions of the articles, and articles were retained if they met inclusion criteria. When the results were inconsistent, the discrepancy was resolved through either discussion or by the introduction of a third rater (Q.X.). Risk of bias was assessed by two reviewers using the Effective Public Health Practice Project’s (EPHPP) Quality Assessment Tool for Quantitative Studies, which includes eight components (21 items) [36]. A rating of weak, moderate, or strong was given to each of the first six components, and these scores contributed to a global rating for the study. Qualitative data was assessed by the Critical Appraisal Skills Programme (CASP) checklist [37].

Data extracted from included studies comprised: authors and date of study, type of literature, where the research design was to be carried out, research methods, data sources, procurement lots, selected drug types, outcome indicators, and empirical results. All data required to answer the study questions were published within the papers, so no contact with authors was necessary.

### 2.3. Data Analysis

The data were analyzed using a range of methods (1) Note Express v3.0 (http://www.inoteexpress.com/aegean/, accessed on 15 October 2021) was used to quantitatively analyze characteristics of the literature such as publication date, keywords, and source. (2) Excel 2016 (Microsoft Corp., Redmond, WA, USA, accessed on 20 October 2021) was used to identify the effects and problems mentioned in the literature, carry out extraction, classification, and similarity merging, and count the frequency of occurrence of different factors. The analysts distilled and coded all the evaluation factors involved in the quantitative/qualitative study. Positive impacts and benefits were coded as A (A1, A2, …, An), while negative feedback and risk factors were coded as B (B1, B2, …, Bn). In this way, all evaluation factors and their frequencies were obtained, which paved the way for subsequent evaluation based on stakeholder classification and health impact dimensional classification, presented via radar charts. (3) Finally, IBM SPSS v23.0 (IBM Corp., Armonk, NY, USA, accessed on 29 October 2021) was used to conduct multidimensional scaling model analysis and transform the co-occurrence matrix of problems into visualized results to enable us to understand their absolute and relative key distributions and degrees of association.

#### 2.3.1. Stakeholder Analysis

Stakeholders in the context of VBP and the drug supply chain include drug manufacturers, drug distributors, retail pharmacies, bidding and procurement agencies, health insurance agencies, public hospitals, doctors, and patients. Based on the availability of documentary evidence, the following five main stakeholders were identified in this study: health insurance management agencies, public hospitals, doctors, patients, and pharmaceutical companies.

Health insurance management agencies play a role in strategic purchasing, adjusting the income and expenditure of health insurance funds through VBP, and managing the mode and implementation of hospital procurement. With regard to public hospitals, VBP promotes the reform of public hospitals, while influencing hospital performance and drug use. Doctors are the direct handlers of drugs and prescriptions. Patients, who represent the demand side of healthcare services, are the biggest beneficiaries of the policy. Pharmaceutical companies are the market participants in VBP, and thus the bidding results directly influence their survival and development.

All evaluation factors extracted have an impact on one or more subjects and invoke positive and negative differences. The extracted evaluation factors were assigned to each stakeholder. The frequency of each perspective for each subject was calculated to obtain the total frequency. The degree of significance was indicated by the collection and proportion of multiple factors. The more the factors, the more reliable the result [38]. A higher frequency represented a greater degree of social concern and strength of actual argument. The proportion is the stakeholder benefits (or risks) expressed as a fraction of the total evaluation factors. It should be noted that some evaluation factors involved multiple stakeholders, and so there were some overlaps.

#### 2.3.2. Health Impact Assessment

Health impact assessment (HIA) is an integrated set of procedures, methods, and tools for assessing the potential impacts of a non-healthcare intervention (policy, plan, or project) on the health of a specific population and its distribution throughout that population [18,39]. It involves multiple dimensions, including biometric identification, individual behavior, economic background, and natural environment [40]. The nature of HIA is to start from the health-impacting cause or “source” intervention. It creates preconditions for the occurrence “process” and the production of active health intervention “result”, thereby promoting public health. Thus, HIA is valuable for mitigating health risks and promoting health gains [41].

Drug evaluation involves safety, effectiveness, economics, innovation, suitability, and accessibility. Based on the treatment purpose of the drug, three evaluation dimensions (drug accessibility, drug-use rationality, and drug quality) were selected by coordinating the HIA goals of health gain and sustainable development. Drug accessibility includes the affordability and accessibility of drugs and drug-use rationality is indicated by the intensity of drug use, while drug quality includes drug effectiveness and safety. Using HIA to evaluate VBP overcomes the current evaluation system that ignores the external environment and human factors, while also raising public awareness and catering to the targets of “National Health”.

As with the stakeholder model evaluation method, the evaluation factors are grouped into each dimension, differentiated according to positive and negative perspectives, to obtain the corresponding frequency totals and frequencies. The benefits and risks of each dimension are thus compared.

#### 2.3.3. Multidimensional Scaling

MDS, a commonly used tool for dimensionality reduction and visualization analysis of complex data, is mainly used to test the interrelationships among various items [42]. MDS has mostly been used in the fields of psychology, behavioral cognition, and sociology [43]. The principle underlying MDS is the use of statistical scores from a set of items or different indirectly measured results as data input, with the relationships among the items displayed on the coordinate axes; similar items are close together, while distinct items are farther apart [44]. The content located at the center of the coordinate system is related to more points and is more crucial, while the content located at the periphery has a weaker relationship with other points [2,45]. The fit of the model was tested using stress (calculated based on the Kruskal stress value) and Dispersion Accounted For (DAF, equivalent to the squared correlation—RSQ of classic MDS). If stress is ≥0.2, the fit is poor; if it is ≤0.1, the fit is satisfactory; if it is ≤0.05, the fit is good; if it is ≤0.025, the fit is excellent; and if it is 0, the fit is ideal, indicating a complete match. The RSQ value is generally acceptable if it is >0.6, and the closer to 1 the better.

MDS was used to summarize the associations between the identified problems and to reveal the key dimensions. In this study, MDS synthesized only the negative evaluation factors for each type of subject and dimension. Instances when the same indicator appeared together in more than one document were shown as a co-occurrence matrix, then imported into the SPSS software program and transformed into a similarity matrix, followed by MDS analysis.

## 3. Results

A total of 4622 publications were identified through the electronic database search, of which 333 were initially selected after removing duplications and reading their titles and abstracts. Sixty-one publications were selected based on the inclusion and exclusion criteria after screening their full texts. A further 18 documents were obtained through an online search and internal resource supplementation. Therefore, a total of 79 documents were included in this study. The selection process is shown in Figure 1. See Appendix A for details of each literature report.

### 3.1. Literature Summary

The main analysis in this study was based on 61 published studies, and 18 supplementary documents were used to support subsequent analysis. Fifty-nine of the 61 documents identified through the database search were journal articles and two were theses. Fifty-six were published in Chinese and five in English. The number of publications increased with time: 10 documents were published in 2019, 18 in 2020, and 33 in 2021. From the perspective of study design, quantitative studies were most common (32), while there were 22 qualitative studies and seven with a mixed study design. The common indicators for quantitative studies included total drug consumption, purchase volume, defined daily dose, frequency of drug use, and daily drug cost.

Some studies (26.2%) analyzed drug use and patient burden in relation to specific diseases, while others (32.8%) compared the changes in drug costs or healthcare costs for patients before and after the implementation of the policy. Other studies examined the effects of the VBP policy based on medication replacement rate, mean cost per outpatient and emergency visit, outpatient drug withdrawal rate, purchase order execution rate, and medical facility payback rates.

In terms of the VBP round evaluated, 68.9% of the studies focused on drugs selected in the first round (including the “4 + 7” pilot cities). Regarding the drugs evaluated, most studies focused on cardiovascular drugs, psychiatric drugs, antibiotics, and anti-hepatitis-B drugs.

### 3.2. Evaluation Factor Extraction

22 positive and 36 negative feedback indicators were extracted as factors of policy effects, as shown in Table 1. It should be noted that only factors with a frequency of occurrence ≥ 2 are listed, and these are ranked from highest to lowest frequency. Some evaluation factors associated with improved patient satisfaction, difficulties associated with public hospital operations, and attenuation of doctor–patient conflicts are not listed because they were rarely mentioned in the selected studies.

### 3.3. Comparison of Stakeholders

The evaluation factors were classified in terms of the five selected stakeholders. Two perspectives are indicated for each stakeholder: benefits and risks (see Table 2). Here, the total frequency was 416, which was obtained by totaling the frequencies of evaluation factors without repetition in Table 2.

Figure 2 shows that patients experienced the most significant effects as a result of the VBP policy. Overall, the benefits for patients were greater than the risks, and the key factors were the accessibility, effectiveness, and safety of drugs (including selected drugs, original drugs, non-selected drugs, and low-priced drugs). However, the risks were greater than the benefits for health insurance management agencies, pharmaceutical companies, and doctors. There were few policy benefits for the doctors. The benefits and risks for healthcare providers were close, with the main benefits being safety in the use of medicines and ease of hospital reform and transformation, and the main risk was that the system of indicators does not reflect the actual needs of the hospital.

### 3.4. Health Impact Assessment

The evaluation factors were classified and included based on the three health impact dimensions: drug accessibility, drug-use rationality, and drug quality. Each dimension involved two perspectives: effects and problems. It should be noted that the factors used for HIA were only some of the factors listed in Table 1, and the dimensions did not overlap. Therefore, the total frequency (233) was obtained by adding the frequencies of the evaluation factors. The classification is summarized in Table 3.

Figure 3 shows the proportions of the effects and problems across the three dimensions. The benefits of accessibility, or the higher level of social acceptance, outweighed its negative effects. In terms of drug quality, effectiveness and problems were very close. Regarding medication rationalization, the positive effects slightly outweighed the problems.

### 3.5. Commonality Analysis of the Problems

The 36 problems (B1, B2, …, B32) that were identified among the evaluation factors related to VBP were screened to extract a co-occurrence matrix (see Figure 4). The values on the diagonal of the matrix represent the frequency of occurrence of the problems mentioned in a given document, while the horizontal and vertical values represent the number of co-occurrences of two problems described in the same document. Given that the standards used to measure the problems differed among the documents, we used the MDS model based on optimal scale transformation (PROXSCAL), and used similarity analysis to identify the correlations between problems. The results showed that stress = 0.1968 and DAF = 0.9146. The fit was average in terms of the stress value, but ideal based on the RSQ value.

Figure 5 shows the results for VBP problems. Overall, the 32 problems were relatively distant and uniformly distributed across the four quadrants without forming an overall cluster, indicating that VBP has various limitations. Based on the position of the coordinates and the distance between two points, there were three main findings:(1)“Excessive price reduction results in the suspension of drug supply, affecting the continuity of drug use” (B1), “The prescription rights of doctors are limited and their response is not positive” (B4), and “Fixed drug use replaces the scientificity of rational drug use” (B19) were located at the center of the coordinate system. This indicates that these three problems were relatively prominent, and the other problems were all related to them.(2)“The current bidding evaluation standards lack scientificity and rationality, and cannot accurately indicate drug quality” (B9) and “The purchase volume standards are unrealistic due to false reporting by hospitals” (B17) were located closest to each other and relatively close to the center. This is indicative of the association between the bid evaluation standards and the purchase volume standards, with similar overall significance.(3)“The consumption of key monitored varieties is accelerated, or there may be problems such as antibiotic abuse” (B23), “Unselected companies are forced to withdraw, leading to a market monopoly” (B10), and “Linked price cuts reduce profits, and the enthusiasm of companies for research and development is weakened” (B34) were located at the edge of the grid. Therefore, these three problems were not closely associated with other problems.

Based on the degree of aggregation of the problem distribution, four clusters can be roughly identified:(1)The first quadrant mainly involved the risks of rising health expenditure and increased fund expenditure. After implementing VBP, non-drug costs increased and the overall cost control effect was insignificant. In the long run, companies colluded to raise prices, placing increased pressure on health insurance funds.(2)The second quadrant mainly highlighted the accessibility of drugs and problems with the rationality of the system design. Accessibility issues were manifested in the fact that the supply of drugs passing the consistency evaluation procedure was insufficient, their varieties and dosage forms were incomplete, the use of original drugs decreased in public hospitals, and the supply of cheap drugs led to decreased drug-use selection among patients and insufficient production capacity among manufacturing companies. Problems with the rationality of the system design were demonstrated by the fact that regional price differences were unreasonable, the role of hospitals in negotiated purchasing was weakened, the prescription rights of doctors were limited, and the fixed drug catalog ignored individual differences in patients.(3)The third quadrant comprised policy fairness and moral hazards. With regard to policy fairness, fewer drugs were used for certain populations such as women and children, treatment coverage was limited, drug availability in rural areas was lower than in cities, and the bid evaluation standards and pricing strategies were unreasonable. As for moral hazards, these were manifested in the fact that hospitals experienced problems relating to false reporting and delayed payment collection, while companies sourced new raw materials to reduce costs, leading to bribery and rent-seeking behavior.(4)The fourth quadrant comprised drug effectiveness and safety. This included a higher incidence of adverse reactions, low levels of doctor–patient recognition, lower quality standards for generic drugs, and discrepancies in terms of the quality and efficacy of different drug varieties.

## 4. Discussion

This study aimed to present comprehensive evidence in terms of various stakeholders and the health impact dimensions of drugs. The policy has not yet balanced the various stakeholders, which is consistent with the difficulty of balancing stakeholder interests in the supply of essential medicines in a global context [46]. In the case of health insurance management agencies, policy design considerations were insufficient, and the cost control and quality guarantee goals were not achieved, triggering profit-seeking behavior by some subjects. Consequently, the effect of health insurance management agencies on optimizing the pharmaceutical market was reduced. The overall profit of pharmaceutical companies was reduced, and there were difficulties in collecting payments, which influenced their enthusiasm for production and motivation for research, development, and quality assurance. As for doctors, their role in rational drug use and their prescription rights were impacted, and there were almost no policy benefits.

In terms of the health impact dimensions, drug accessibility was the area of greatest concern. One reason for this is that the measurable indicators and data sources regarding accessibility were relatively clear, and many studies assessed drug accessibility in terms of varieties provided and changes in cost. Another reason is that the prices of selected and non-selected drugs showed remarkable variation, with increased affordability. However, the disrupted supply of various drugs led to poor drug accessibility. The problems regarding drug-use rationality and drug quality were greater than the positive effects. The lack of sufficient empirical data resulted in significant concerns among all parties. The largest problem regarding drug quality was that the profits of companies were reduced, even though their production costs decreased. Drug-use rationality was influenced by drug quality, and there were potential risks of antibiotic abuse and higher drug use in clinical applications.

In summary, VBP has been effective in addressing poor access to essential medicines, saving healthcare costs and accelerating generic substitution, which is consistent with the results of studies by Peivand et al. in Iran [2] and Chaumont et al. in Mexico [47]. However, it still lacks guarantees about the sustainability of the fund, balancing the needs of market players, and constructing a reasonable indicator system to ensure the safety of clinical drug use, which is consistent with the results of studies by Dylst et al. in Europe and Roy et al. in Delhi [28,48]. The World Health Organisation made a broad statement that inefficiencies in health systems in low- and middle-income countries are around 20–40% [49]. The reallocation of resources within the health system in China will inevitably give rise to a series of problems. Combined with our MDS results, the clustering of issues can be clearly observed, and in order of urgent importance the main issues focus on the following areas:

First, drug quality has yet to be confirmed. At present, the main focus of centralized drug procurement in China is on generic drugs and original drugs that have exceeded their patent term. These drugs can be included in VBP only after passing the necessary quality and efficacy consistency tests. However, after taking the risk factors of drug use into account, the effectiveness and safety of the drugs selected for the VBP program have been questioned by doctors as well as patients. Some studies have also found that the replacement of original drugs by generic drugs leads to reduced clinical effects and patient compliance [49,50,51]. Doctors have doubts about the quality of generic drugs, like the physician from Ohio who recognized that the Food and Drug Administration(FDA) is unable or unwilling to ensure that the quality of all marketed generic drugs is consistent [52].

To further verify the efficacy and safety of the generic drugs selected for VBP, the National Healthcare Security Administration conducted a study of clinical efficacy and safety using data from several public hospitals relating to 14 drugs included in the first round of VBP. The results showed that “the selected generic drugs were consistent with the original drugs in their pharmaceutical composition and were bioequivalent, which realized clinical equivalence and truly achieved price reduction without quality decline” [53]. Questions that require further consideration include how to guarantee the quality of all selected drug varieties and how to use results from evidence-based medicine to strengthen support. There are also international calls for the use of real-world data to ensure the clinical equivalence of generic drugs [54].

Second, the level of satisfaction of the clinical needs of doctors is low and their prescription rights are limited. In the process of clinical drug use, because doctors are aware of a patient’s various conditions, if they can rationally use drugs based on individual differences among patients, adverse drug reactions can be greatly reduced. In the context of VBP, doctors obtain few benefits compared with other stakeholders, and yet their prescription rights are restricted. This conflicts somewhat with the findings of Han [55] and Wang [56], who concluded that restricting prescription behavior promotes rational drug use and effectively controls drug expenditure. However, rigid administrative regulations can affect doctors’ and patients’ options in relation to drug use, and thus their level of satisfaction with the policy. Moreover, the reduction in drug prices means that doctors’ incomes are significantly reduced, while the new compensation mechanism is not yet complete. This has influenced doctors’ enthusiasm for prescriptions, which has had an unfavorable effect in terms of achieving “health insurance cost control” and “rational drug use.”

In the diagrammatic representation of the MDS results, this problem is also located in a central position. Thus, improving the policy recognition and prescription enthusiasm of doctors will be beneficial for promoting rational drug use and facilitating hospital value transformation.

Third, even though the short-term target of price reduction has been reached, there are numerous long-term risks. With regard to drug distribution, the supply side proposes to focus on obtaining low prices, while the demand side pursues the rational and economical use of drugs [57,58]. From the perspectives of patients and health insurance management agencies, VBP has basically achieved the following goals: the prices of selected drugs have been substantially reduced, and both the mean drug cost and the healthcare cost per patient visit have decreased to a certain extent. Meanwhile, the “spillover effect” of the policy has resulted in a decrease in the consumption and costs of other drugs [59]. The sharp decrease in the mean cost per patient visit has resulted for patients in a simultaneous improvement in the affordability and accessibility of basic drugs and other innovative drugs.

However, although VBP has significantly reduced the prices of selected drugs in the short term, healthcare costs, including drug costs, are still on the rise, and health insurance fund expenditure is also at risk of rising. Furthermore, follow-up development might be weak because of conflicts between policy fairness, standard scientificity, and other policies. Hence, the effects and risks of the policy need to be evaluated over a longer period.

Finally, the disruption of the drug supply chain has initiated discussions on drug costs and prices. As a result of VBP, patients receive the direct benefits of the price reduction policy, but the profits of selected companies are significantly reduced. With the expansion in the range of drugs purchased, these companies face pressure not only to achieve the required purchase volume, but also to deal with the risks of changes in production costs. Nevertheless, the purpose of VBP is to eliminate the excess costs that have long existed in the field of drug distribution, rather than to reduce the production costs of companies. Participating companies should not take the approach of submitting quotes that are below the cost of production.

Drug supply disruption not only affects treatment effectiveness, cure rate, and drug-use safety among patients, but may also exacerbate conflicts between doctors and patients, as well as between the government and companies [60]. Thus, it is necessary to consider establishing an emergency reserve and shutdown reporting system for production enterprises, and to adopt disciplinary measures in response to actions that fail to guarantee supply [61].

Looking at the goals of the national policy over the past three years, our results indicate that overall, the four main goals—price reduction and quality assurance of drugs, transformation and upgrading of the pharmaceutical industry, further reform of public hospitals, and burden alleviation and efficiency improvements in healthcare security—have gradually been achieved. Meanwhile, there are several aspects of VBP in China that need to be improved.

Life-cycle management of drug purchasing should be implemented. A complete purchasing mechanism is required before VBP drugs enter the distribution network. An effective drug purchasing process guarantees that the correct drugs are provided at reasonable volumes and prices, and meet recognized quality standards. Otherwise, various problems can occur, such as drug shortages, overstocking, resource wastage, and purchase of low-quality products [62]. The “Operational Principles for Good Pharmaceutical Procurement” released by the World Health Organization state that government departments and public hospitals should follow four basic operational principles when implementing the purchasing process: first, clear, effective, and transparent management, purchasing functions and responsibilities for relevant departments such as selection, quantification, product specifications, pre-selection of suppliers, standardized purchasing procedures, proper planning, and regular monitoring; second, reasonable drug selection and volume determination, whereby the order volume should be based on reliable estimates of actual demand; third, arrangement of financing and competitive safeguards, and maintenance of market fairness; and fourth, supplier selection and quality assurance [63].

Reasonably formulated indicators and positive incentive mechanisms should be established. Agreed purchase volume is the main differentiating feature of VBP from the previous method of purchasing by invitation to bid, and thus it is important to manage the decoupling of volume and price. Presently, total purchase volume is based on 50–80% of annual drug consumption, and specific purchase volume is associated with the number of selected companies [64]. However, specific purchase volume should not only consider enabling the survival of unselected companies, but also take into account the operational attributes of hospitals and the drug-use characteristics of different departments throughout the year.

Furthermore, hospital performance appraisal should focus not only on the success of purchasing methods, but also on the correct use of drugs [65]. Prioritizing rational drug use, doctors should be guided step-by-step toward using the selected drugs. Meanwhile, the supply of original drugs should be guaranteed to meet the needs of patients and doctors. Then, by linking health insurance payment rules with doctor evaluation systems, positive incentive mechanisms can be gradually established to increase the enthusiasm of doctors.

Continuous monitoring of drug use and comprehensive clinical evaluation should be implemented. As VBP continues to expand in terms of area and coverage, the proportion of selected drugs in the drug catalog of public hospitals will continue to increase. Therefore, it is necessary to carry out research on a regular basis. The quality of and any adverse reactions to selected products must be continuously monitored, and close attention should be paid to drugs such as antibiotics [66]. Furthermore, corresponding treatment regimens should be formulated.

Monitoring the clinical use of VBP drugs is beneficial for conducting comprehensive evaluations of their safety, effectiveness, and economics [67]. The Circular of the General Office of the National Health Commission issued a Notice on Standardizing the Work of Clinical Comprehensive Evaluation of Drugs (2021) No. 16, which stated that drug-use monitoring and comprehensive clinical evaluation should be carried out to provide evidence for optimizing the drug supply catalog, promote the rational clinical use of drugs, control unreasonable expenditure on drugs, and ensure timely payment collection [68].

The government’s position should be clarified and policy linkage strengthened. The government is the driving force behind policy implementation. Thus, it should adopt appropriate policy designs and efficient supervision methods to ensure the realization of its policy goals. The first task is to improve VBP-related policies and strengthen the integrity and cooperativity of various healthcare reform policies, including drug price determination mechanisms, health insurance payment standards, and generic drug regulations. The next task is to clarify its role and position, to explore fair treatment methods that prevent the government from being both a “referee” and a “player”, and suppress rent-seeking corruption. Alternatively, based on Shenzhen’s examination of the GPO model in the United States, a third-party purchasing organization could be entrusted with this role to prevent the emergence of an administrative monopoly.

## 5. Contributions and Limitations

We declare that this research is published here for the first time worldwide. This study is the first to integrate all the factors used to evaluate the effects of VBP policy and to measure the policy’s benefits and risks from two different perspectives, namely, stakeholders and health impacts, which is in line with the healthy and stable development of the market. The MDS used in this study provides clear evidence of similarities and associations but has rarely been used for policy evaluation. In the future, consideration should be given to applying this method to the field of policy decision-making, as it is a useful tool for clarifying the relationships among various phenomena and determining the degree of importance and urgency. However, this study also has some limitations. First, we did not conduct a thorough evaluation of the quality of evidence in each of the 79 documents included in our sample, although this would be consistent with the purpose of the study, i.e., to find in a range of documents evidence of the positive and negative impacts of VBP policy, rather than determining which of the documents presented more or less reliable evidence. Second, some differences were ignored when summarizing each factor. For example, descriptions of cost reductions for patients with different indications such as either “slightly reduced” or “remarkably reduced” were collectively classified as “cost reduction,” which might have resulted in an overestimation of the positive effects of the policy. Third, in this study we transformed the findings of qualitative studies into quantitative results. MDS analysis provides a visual representation that reduces potentially complex datasets to their main dimensions, and the interpretation of the results is necessarily based on one’s own experience. Thus, there was a certain degree of subjectivity involved in our interpretation of the results, and subjective understanding was inevitable in relation to determining the number of dimensions, interpreting the main dimensions, and classifying the items. Fourth, this study did not identify any causal relationships between various factors, an aspect that increases the difficulty of decision-making and requires further research.

## 6. Conclusions

VBP is a focus of conflict in the healthcare industry in China, and our study assesses it from two different perspectives: stakeholder and health impact, in line with the need for market equilibrium and health value orientation. The results show that the interests, values and attitudes of policy makers and recipients have not yet been balanced. Although the ideal end state of drug procurement has not been reached, expectations have been met in terms of reducing the burden on patients and reducing fund expenditure. The key areas for improvement are safeguarding the quality of drugs, setting procurement targets differently, and promoting rational drug use. VBP has transformed from an initial pilot program to a fully advanced system. In the follow-up promotional process, it is necessary to emphasize the unshakable nature of the health benefit goal, taking into account the interests of all parties, and to eliminate some of the negative factors in order to optimize policy design. In the long term, a new system of value-based payment should be established in relation to the healthcare system, and outdated drug production methods should be eliminated, enabling China to achieve the ultimate goals of meeting clinical needs, reducing patient burden, rationalizing industry layout, and improving system efficiency.

## Figures and Tables

**Figure 1 ijerph-19-04285-f001:**
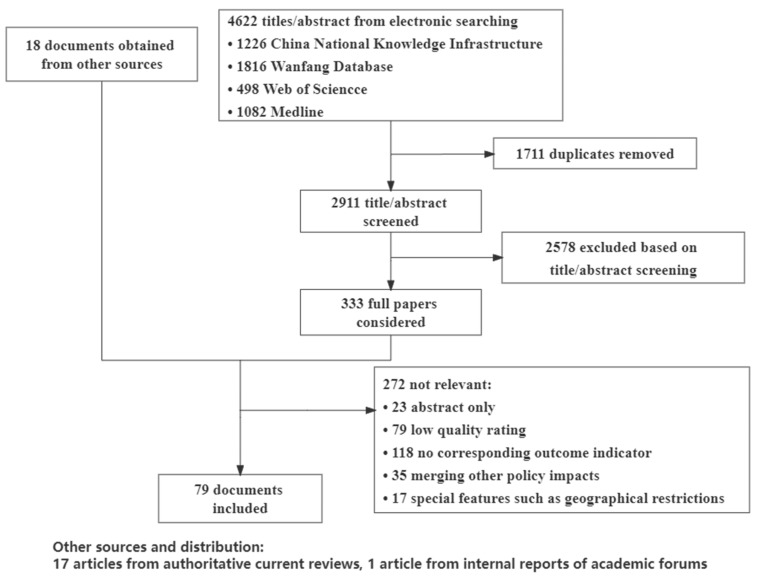
PRISMA flow chart for inclusion/exclusion of documents.

**Figure 2 ijerph-19-04285-f002:**
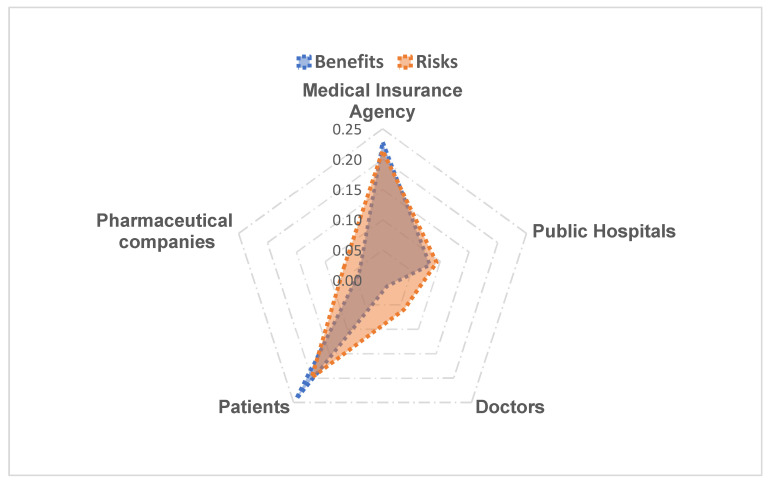
Radar chart of benefits and risks for various stakeholders.

**Figure 3 ijerph-19-04285-f003:**
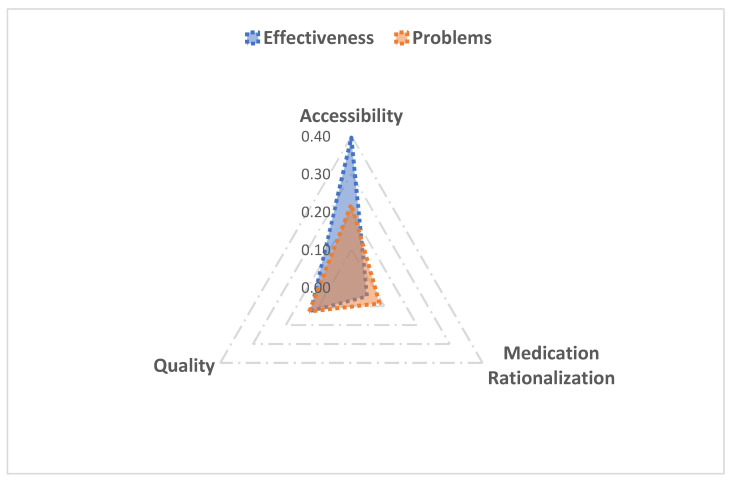
Radar chart of effectiveness and problems of health impact evaluation dimensions.

**Figure 4 ijerph-19-04285-f004:**
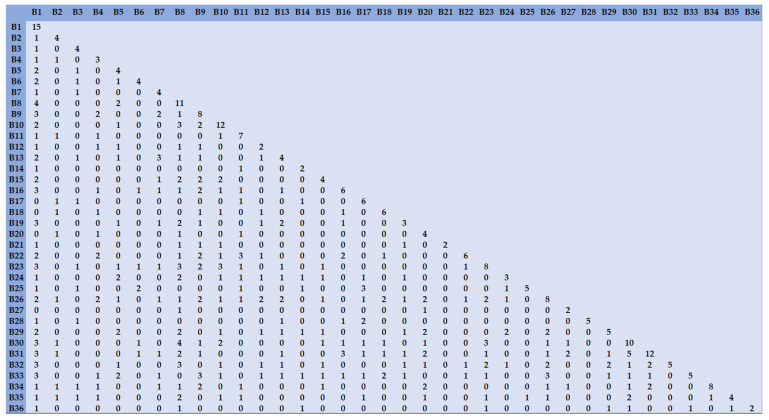
The 36-question co-occurrence matrix.

**Figure 5 ijerph-19-04285-f005:**
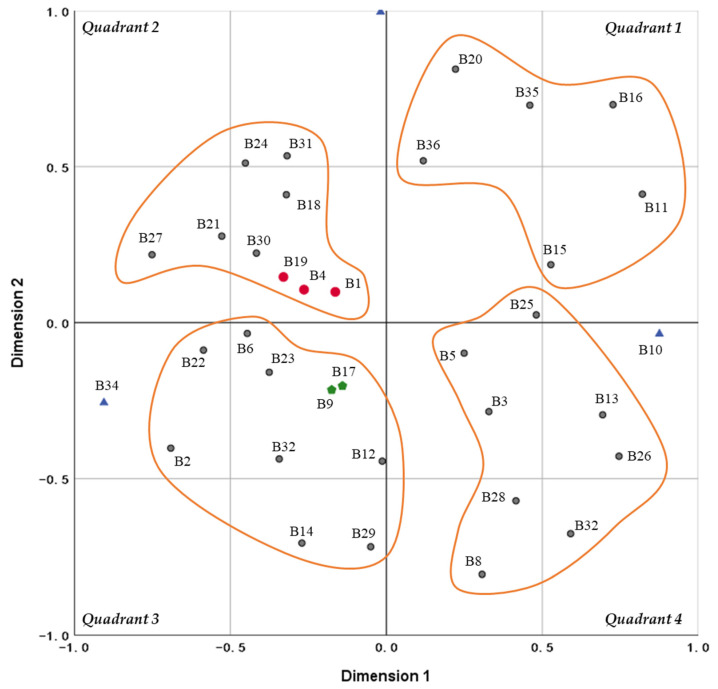
Multi-dimensional scale analysis coordinate map.

**Table 1 ijerph-19-04285-t001:** Evaluation factors regarding the effect of volume-based procurement (VBP) of drugs.

Classification	Description	Frequency	Code
Effects	The price of selected drug varieties drops and the mean cost per patient visit declines	52	A1
	It saves health insurance funds and gives room for more innovative drugs	21	A2
	The distribution links of drugs are optimized	16	A3
	The price of unselected varieties falls gradually, and the policy spillover effect is remarkable	14	A4
	Promoting industry merger and reorganization, and forcing enterprise innovation	14	A5
	Independent quotation for companies saves marketing expenses, reduces transaction costs	11	A6
	The monitoring work of adverse reactions is stable, with a low number of reports	9	A7
	The effectiveness and safety of selected drugs are consistent with those of original varieties	9	A8
	It is the entry point for strengthening medical reform and promoting three-medicine linkage	9	A9
	The prescription behavior of doctors is standardized and rational drug use is guided	7	A10
	The expected diagnosis and treatment effects can be achieved with low dressing change rate and good compliance	6	A11
	The global budget system concentrates funds, shortens enterprise capital turnover	6	A12
	The proportion of hospital drugs declines, which forces public hospitals to provide value-oriented services	6	A13
	The shortlisted item passes the consistency evaluation and has a quality improvement effect	5	A14
	It gives full play to the decisive role of the market in allocating resources	5	A15
	Balance sharing within the framework of global budgets can be used for salary system reform in the long term	5	A16
	The consumption of selected drugs and original substitutes is increased, which optimizes the drug catalog	4	A17
	The selected varieties have sufficient supply and timely delivery	3	A18
	The selected varieties are focused on common diseases, with a wide range of beneficiaries	3	A19
	Patient demand for drug use is released	3	A20
	The consistency of drug use within medical consortia is guaranteed	3	A21
	The supply of essential drugs is safeguarded	2	A22
Problems	Excessive price reduction results in the disruption of drug supply, affecting the continuity of drug use	15	B1
	The coverage of selected drugs is limited	12	B2
	Patients have low acceptance	12	B3
	The prescription rights of doctors are limited and their response is not positive	11	B4
	The recognition degree of doctors is low and they tend to use original drugs in sensitive fields	10	B5
	The standards for agreed purchase volume of selected drugs lag behind, and the indicators of different departments are unreasonable	8	B6
	There is a problem of raw material replacement, with doubts about drug effectiveness and safety	8	B7
	There exist differences in the quality and efficacy of different drug varieties	8	B8
	The current bid evaluation standards cannot accurately indicate drug quality	8	B9
	Unselected companies are forced to withdraw, leading to a market monopoly	7	B10
	Health insurance fund expenditures are at risk of increasing in the long term	6	B11
	The delivery rate is low and delivery is delayed at the grassroot level	6	B12
	The price of non-selected common drugs in social pharmacies and through other channels is rising	6	B13
	Defaults on payment are serious, causing a moral hazard	6	B14
	Dosage is increased in clinical use to achieve the drug’s effect	5	B15
	The overall cost control effect is not remarkable	5	B16
	The purchase volume standards are unrealistic due to false reporting by hospitals	5	B17
	The selected prices are relatively high in areas with limited health insurance funds, while low prices are seen in areas with sufficient funds, forming an “upside down” pattern	5	B18
	Fixed drug use replaces the scientificity of rational drug use	5	B19
	Non-drug healthcare costs are increased	4	B20
	Public hospitals suspend the supply of original drugs to reach the agreed consumption of selected drugs	4	B21
	It fails to take care of older people, women, children, and patients with special diseases	4	B22
	The consumption of key monitored varieties is accelerated, or there may be problems such as antibiotic abuse	4	B23
	The drug varieties and dosage forms are incomplete, causing inconvenience for administration by patients	4	B24
	Domestic generic drug standards are lower than international standards	4	B25
	Due to the indicator limitation of tertiary hospitals, there is a lack of motivation to refer patients to lower-level hospitals, affecting the advancement of the hierarchical diagnosis and treatment system	4	B26
	The supply of cheap drugs is disrupted	4	B27
	Drugs of the same specification are supplied at multiple prices, and the prices of drugs with the same generic name are considerably different	4	B28
	Excessive administrative intervention affects resource allocation, leading to rent-seeking behaviors and causing unfair competition	3	B29
	There is a gap between the production capacity of companies submitted for approval in the consistency evaluation and their actual production capacity, leading to weak production.	3	B30
	The bargaining power of hospitals is weakened	3	B31
	The frequency of allergic symptoms with some selected drugs is higher	2	B32
	The strategy of taking only low prices deliberately distorts drug prices and reverses resource allocation	2	B33
	Linked price cuts reduce the enthusiasm of companies for research and development	2	B34
	Selected companies are conspiring to increase drug prices in the long run	2	B35
	Original drugs and biosimilars cannot be replaced horizontally owing to the unique complex spatial structure of biosimilars	2	B36

**Table 2 ijerph-19-04285-t002:** Classification of the evaluation factors based on the perspectives of the five stakeholders.

Stakeholder	Perspective	Factors	Total Frequency	Proportion
Health insurance management agencies	Benefits	A2, A3, A4, A5, A9, A13, A14, A15, A16	95	0.2284
	Risks	B3, B5, B6, B8, B9, B11, B16, B18, B22, B23, B25, B26, B29, B33, B34, B35, B36	89	0.2139
Public hospitals	Benefits	A7, A10, A13, A16, A17, A21	34	0.0817
	Risks	B3, B6, B14, B17, B19, B31	39	0.0938
Doctors	Benefits	A16	5	0.0120
	Risks	B4, B15, B19, B23	25	0.0601
Patients	Benefits	A1, A4, A7, A8, A11, A18, A19, A20, A22	101	0.2428
	Risks	B1, B2, B7, B12, B13, B15, B19, B20, B21, B23, B24, B27, B28, B32	83	0.1995
Pharmaceutical companies	Benefits	A6, A12	17	0.0409
	Risks	B9, B10, B14, B30, B34, B35	28	0.0673

**Table 3 ijerph-19-04285-t003:** Classification of factors in the three dimensions of health impact assessment.

Dimension	Perspective	Factors	Total Frequency	Proportion
Drug accessibility	Effects	A1, A2, A4, A18, A22	92	0.3948
	Problems	B1, B2, B12, B13, B20, B21, B27	51	0.2189
Drug use rationality	Effects	A10, A17	11	0.0472
	Problems	B15, B19, B23, B24, B36	20	0.0858
Drug quality	Effects	A7, A8, A11, A14	29	0.1245
	Problems	B7, B8, B9, B25, B32	30	0.1288

## Data Availability

Data sharing does not apply to this article as no datasets were generated during the current study.

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
