# Peer review of "Has the Volume-Based Drug Purchasing Approach Achieved Equilibrium among Various Stakeholders? Evidence from China"

_ijerph, 2022, doi:10.3390/ijerph19074285_

Round 1

Reviewer 1 Report

I understand and accept the explanations given by the authors to my main criticism regarding the lack of comparison with the international programs of "managed entry access" or "risk-sharing agreements". In view of the modifications made, at the suggestion of other reviewers, I understand that the article may be accepted for publication.

However, it should be improved with some reference for international comparison with other programs that, like VBP, are oriented towards essential medicines with high clinical use, wide coverage and low prices. Certainly, developing countries will have had the same type of problems and therefore will have proposed similar programs. Even countries with more advanced drug access regulation systems had to address them in the past. In this sense, it would be interesting for the introduction to compare the main characteristics of the VBP program with others and for the discussion, if possible, to compare the findings obtained with the results in other countries.

Author Response

These comments make us realise that we should look more at the global international picture and not limit ourselves to Chinese practice. Revised portion are tracked in the paper. The main corrections in the paper and the responds to the reviewer’s comments are as flowing(Details please see the attachment.):

  1. We have mentioned in the paper that the rationale for VBP implementation is similar to the GPO model in the US, Singapore and other countries, and that similar issues have arisen in implementation. This was expressed in the text as:

“GPOs are mainly engaged in competitive bidding and supply chain management for drug procurement, driving effective service provision through economies of scale and improved bargaining power. The GPO model is not perfect, and problems such as price competition and increasing coordination costs(e.g. inventory, transport, supervision costs) have occurred in other countries including the United States and Singapore…VBP in China is currently experiencing similar dilemmas.”

  1. While taking into account the comments of other reviewers, we have supplemented the original draft with information on the procurement of medicines in international federations and other countries, and have added a description of the policy objectives, the manner of implementation and the initial results achieved. This is expressed in the text as:

“Indeed, a number of national or international allied organisations have implemented centralised procurement as a means of creating economies of scale, increasing purchasing power and reducing health system costs. For example, the United Nations International Children's Emergency Fund (UNICEF)... In addition, other countries such as the Gulf Cooperation Council, the East African Community, Delhi, India, Brazil, Brazil, the Caribbean, Mexico, are engaged in volume purchasing with the aim of addressing high drug prices and poor access to essential medicines. “

  1. As you suggested, a comparison with the policies and effects of the use of essential medicines for high clinical use, wide coverage and low prices in countries with more advanced regulatory systems for access to medicines has been added.
  • In the introductory section, a comparison of the ways in which such medicines are procured has been added.

Although the countries differ in terms of the type of procurement, lead sector, participants and process, all have contributed to increasing access to medicines. In addition to the banded procurement approach, Spain has ensured the availability of essential medicines with high clinical use through the signing of discounted generic contracts, and the UK through separate bidding for generic medicines.

  • In the discussion section, comparisons of relevant results are supplemented with comparative references.

This was consistent with the difficulty of balancing stakeholder issues in the supply of essential medicines in a global context.

However, it is still lacking in guaranteeing the sustainability of the fund, balancing the needs of market players, and constructing a reasonable indicator system to guarantee the safety of clinical use of drugs,which is consistent with the results of studies by Dylst et al. in Europe and Roy et al. in Delhi.

Doctors do have doubts about the quality of generic drugs, like a physician from Ohio recognized that the FDA is unable or unwilling to ensure that the quality of all marketed generic drugs is consistent .

There are also international calls for the use of real-world data to ensure the clinical equivalence of generic drugs.

This manuscript is a resubmission of an earlier submission. The following is a list of the peer review reports and author responses from that submission.

Round 1

Reviewer 1 Report

Dear Authors

This is a very interesting paper that is generally well written and presented. More specific comments are included in the manuscript.

Reviewer 2 Report

From my point of view, the work that is analyzed focuses on an excessively strict bibliographic search. It analyzes only what it calls "volume-based drug purchasing" (VBP) using as search terms "centralized drug purchasing" OR "volume-based procurement" OR "volume-based purchasing". This fact and the exclusive focus on China, that applies with the exclusion criteria of the PRISMA model, makes them leave out of consideration all the extensive existing literature on "managed entry agreements" or "risk-sharing agreements", terms frequently used as equivalents. It is common to classify them into two major categories related to the uncertainty problem they address: 1) financial agreements, usually called price-volume agreements, and 2) pay-for-performance agreements, which take into account the outcomes yielded by the use of the health technology. Precisely in the first of these categories can be included the centralized purchases that are intended to be analyzed. It does not seem appropriate to leave out of the study all the literature on the international experience in this type of agreement to focus exclusively on a specific form that, according to the results obtained, is carried out in a single country. In this same sense, the broadening of the focus of the search could help to give solidity to the grouping of the factors in the dimensions that are analyzed of the Health Impact Assessment.

This broadening of focus should serve to review the evaluation factors of the agreements, differentiated in Effects and Problems, in particular to reduce them by facilitating the grouping of some of them since their number (22 and 36 respectively) may be excessive. It can also help clarify and improve the comparability of the stakeholders considered. Although the article mentions multiple stakeholders, it focuses on five specific ones without sufficiently justifying the choice and the consideration of "public hospitals" is doubtful since it is not clear whether they are incorporated as health managers, in which case they could be addedto the health insurance agencies in a joint category of health administration representatives, or they are considered as clinical managers with what could be grouped together with the doctors.

The discussion of the study could also be enriched for comparison with the international experience derived from broadening the focus of the search carried out. It is striking in this section the emphasis placed on the problems of quality of medicines and equivalence of generics with brand originals. These problems are generally solved by administrative means and by the subjection of manufacturers to technical and bureaucratic controls and not by volume-based pricing systems. This could be talking about some peculiarities of the drug value chain in China, which as presented cannot be outlined.

Since these considerations affect the core of the work and their assumption would mean completely redoing the systematic search and the parameters for the subsequent MDS analysis, I consider that the work should be rejected.

Reviewer 3 Report

The present study evaluates the effects of the VBP policy, taking into account the benefits and risks of this policy. It presents some comprehensive evidence in terms of the stakeholders and the health impact dimensions of drugs. The paper is coherent to the scope of the journal and it could be of interest for international readers. I recommend the acceptance of the manuscript after considering the following points:

  • It seems that the authors did not use software to insert the references in the text. I recommend to check the brackets for each citation (e.g. line 64, 82, 89, 93, 254, 338, 519, 605 etc.).
  • Abstract:
    • The aim of the study and the used methods should be presented in the abstract.
  • Introduction:
    • A short description of Chinese Health Insurance System should be included in this section.
    • Lines 64-67: The national VBP was developed for a few drugs included on the Health Insurance System. This aspect should be mentioned.
    • Lines 69-74: can be deleted. They have a low relevance for the present study.
    • Lines 82-83: “In the past, low-quality production of generic drugs was a serious problem, and approval standards were low.” – a reference is necessary and more information about the relation between quality of drugs and their authorization on the market should be added.
    • Lines 104-105: Did the COVID-19 pandemic influence the category of drugs procured through these VBP sessions?
    • Lines 125-126: The principle of group purchasing organizations (GPOs) should be detailed.
    • Lines 127-128: the authors should define the “coordination cost” term.
    • More data regarding drugs procurement systems from different countries should be presented. I recommend some selective, but not restrictive references:
      • https://www.mdpi.com/1660-4601/17/12/4456
      • https://www.minsalud.gov.co/sites/rid/Lists/BibliotecaDigital/RIDE/VS/MET/lineamientos-resolucion-medicamentos-hepatitisc-2017.pdf
      • https://assets.publishing.service.gov.uk/media/59845568e5274a1707000065/108-Evidence-and-experiences-of-other-countries-health-procurement.pdf
    • The authors should mention the aim of the study and describe its objectives.
  • Materials and methods:
    • Paragraph 2.1.1.: Was literature search conducted only for China? It should be mentioned.
    • Paragraph 2.2.2.: How was the HIA realized?
  • Results:
    • Lines 353-367 are more appropriated in the Discussions section.
    • Lines 371-374 are more appropriated in paragraph 2.2.2.
    • Lines 383-397 are more appropriated in the Discussions section.
    • Lines 422-445 are more appropriated in the Discussions section.
    • Figure 5 should contain the number of each quadrant.
    • Lines 450-482 are more appropriated in the Discussions section.
  • Discussions:
    • In this section authors should discuss their results comparing to other studies. I suggest adding some of this information.
    • Lines 484-489 need a reference.
    • Lines 502-509 need a reference.
    • Lines 460-462: Authors should add information regarding this excess costs. Also, this section requests a reference.
    • Lines 648-649: The citation format is different from other citations.
  • Contributions and Limitations:
    • Lines 651-654: The authors should mention if the present study is the first to be published worldwide or the Republic of China
  • Conclusions:
    • Some information regarding the results and according to the title of the manuscript should be added.
  • References:
    • Should be corrected according the guidelines for Authors of this journal.
    • Ref 3, 9 should be completed with more information.
    • Reference 10 should be changed.